# Vector assembly of colloids on monolayer substrates

Lingxiang Jiang[1], Shenyu Yang[1], Boyce Tsang[2], Mei Tu[1] & Steve Granick[3,4,5]

The key to spontaneous and directed assembly is to encode the desired assembly information to building blocks in a programmable and efficient way. In computer graphics, raster graphics encodes images on a single-pixel level, conferring fine details at the expense of large file sizes, whereas vector graphics encrypts shape information into vectors that allow small file sizes and operational transformations. Here, we adapt this raster/vector concept to a 2D colloidal system and realize 'vector assembly' by manipulating particles on a colloidal monolayer substrate with optical tweezers. In contrast to raster assembly that assigns optical tweezers to each particle, vector assembly requires a minimal number of optical tweezers that allow operations like chain elongation and shortening. This vector approach enables simple uniform particles to form a vast collection of colloidal arenes and colloidenes, the spontaneous dissociation of which is achieved with precision and stage-by-stage complexity by simply removing the optical tweezers.

[1] College of Chemistry and Materials Science, Jinan University, Guangzhou 510632, China. [2] Department of Physics, University of Illinois, Urbana, Illinois 61801, USA. [3] Center for Soft and Living Matter, Institute for Basic Science (IBS), Ulsan 44919, Republic of Korea. [4] Department of Chemistry, UNIST, Ulsan 44919, Republic of Korea. [5] Department of Physics, UNIST, Ulsan, 44919, Republic of Korea. Correspondence and requests for materials should be addressed to L.J. (email: jianglx@jnu.edu.cn) or to S.G. (email: sgranick@ibs.re.kr).

An ongoing endeavour in chemistry and materials science is to assemble building blocks like molecules and colloids into organized structures to ultimately match or even surpass nature's precision, complexity and functionality[1–11]. Central to this endeavour is an elusive task—how to encode the desired assembly information to the building blocks in a programmable and efficient way? Computer graphics, though a seemingly irrelevant field, provides a fresh perspective to tackle this problem. Since its birth in the 1950s, two main techniques have been developed hand-in-hand with one complementary to the other: raster or pixel graphics, in which images are edited on the single-pixel level (Fig. 1a), and vector graphics, which encodes information of shapes and colours into 'vectors'—lines, curves or objects leading through certain pin points (Fig. 1b)[12,13]. While the former can achieve photo-realistic details at the cost of a large file size, the latter is ideal for simple or composite images featuring minimal file sizes and operational transformations such as translation, rotation and scaling.

We propose that the raster/vector concept can be concretized in spontaneous and directed assembly systems. Analogous to computer graphics, raster and vector assembly are defined by their methods of information encodement. Take DNA origami as an example:[14–16] the Rothemund method is a raster-like one that encrypts an enormous amount of folding information into ~200 different oligonucleotides (pixels) via a sophisticated, computer-aided algorithm to produce arbitrary shapes[15]. In comparison, the Dietz-Douglas-Shin method is, in our opinion, a vector-like one that concentrates a minimal amount of assembly information into bend segments (pin points) to form simple geometries[16]. The possible merits of vector assembly therefore prompt us to apply the raster/vector concept to other systems.

In this paper, we focus on a 2D colloidal assembly system in which spherical particles sediment down to a substrate. Optical tweezers[17–19] are employed to encode assembly information to the particles for them to form structures and patterns. In the case of a flat glass substrate, each particle is held by an optical tweezer into position to form, for example, a 10-particle chain and a 4-by-4 square (Fig. 1c,e). This kind of directed assembly as well as most reported optical tweezer systems[18] are raster-like. On the contrary, vector assembly is realized when the flat substrate is replaced by a colloidal monolayer. For example, a zigzag chain and a perylene-like structure are held stable by a minimum number of optical tweezers on the pin points (two ends and four corners, respectively, in Fig. 1d,f). Notably, the vector method can greatly reduce the necessary number of optical tweezers by ~80%. The following paper is organized in such a way that general readers can quickly grasp the raster/vector idea from the Results section while readers interested in experiments can look into the Methods section first.

## Results

**Vector colloidal assembly directed by optical tweezers**. The experimental set-up is briefly described here, while details can be found in Methods. Onto a monolayer of close-packed 3 μm-diameter silica particles (bottom particles), dilute 3.4 μm-diameter silica particles (top particles) sediment due to gravity. The top particles are randomly distributed across the substrate, incapable of forming any ordered structures without optical tweezers (Fig. 2a). Brownian diffusion of the top particles is highly slaved by the corrugated surface of the bottom monolayer (Supplementary Movie 1): they are restricted to basins most of the time and can hop to nearby basins only through valleys but not over hills. Multiple optical tweezers were then employed to manipulate the top particles to form vector structures.

Vector structures were assembled by a minimum number of optical tweezers in a step-by-step iterative manner. The steps to construct a zigzag chain with three optical tweezers is schematically illustrated in Fig. 2b. Step 1, tweezers **1** to **3** are used to move three particles into positions. Step 2, tweezer **2** releases its particle and is relocated to a new particle. Step 3, tweezer **2** fetches the new particle to a desired position. Step 2 and

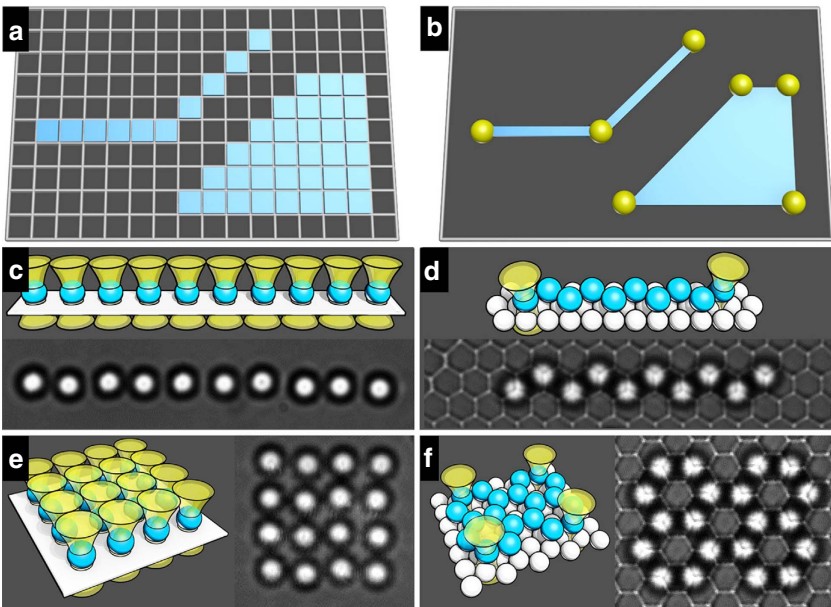

**Figure 1 | The raster/vector concept in computer graphics and assembly.** (**a**) Raster graphics made of pixels. (**b**) Vector graphics highlighting pin points. (**c,e**) Raster assembly of particles, as directed by optical tweezers, on a flat substrate into a straight chain and a square. The assembly information is encoded in 10 and 16 optical tweezers, respectively. (**d,f**) Vector assembly of particles, as directed by optical tweezers, on a colloidal monolayer into a zigzag chain and a perylene-like structure. The assembly information is concentrated into 2 and 4 optical tweezers, respectively. In the schematic illustrations, yellow funnels represent optical tweezers, blue spheres are top particles and white spheres are bottom particles that form the monolayer substrate beforehand. In the optical microscopy images, the spheres are 3.4 μm in diameter.

3 are repeated by employing tweezer **2** or **3** alternatively as the shuttle to add particles to the chain and the other two tweezers as the pins to fix the chain ends. Finally, a 6-particle zigzag chain is formed and stabilized by two tweezers. This iterative procedure can be used to elongate or shorten a chain at will, resembling the scaling operations in vector graphics.

As we discuss in the Methods section, the bottom monolayer and pinning optical tweezers are crucial to the stability of vector

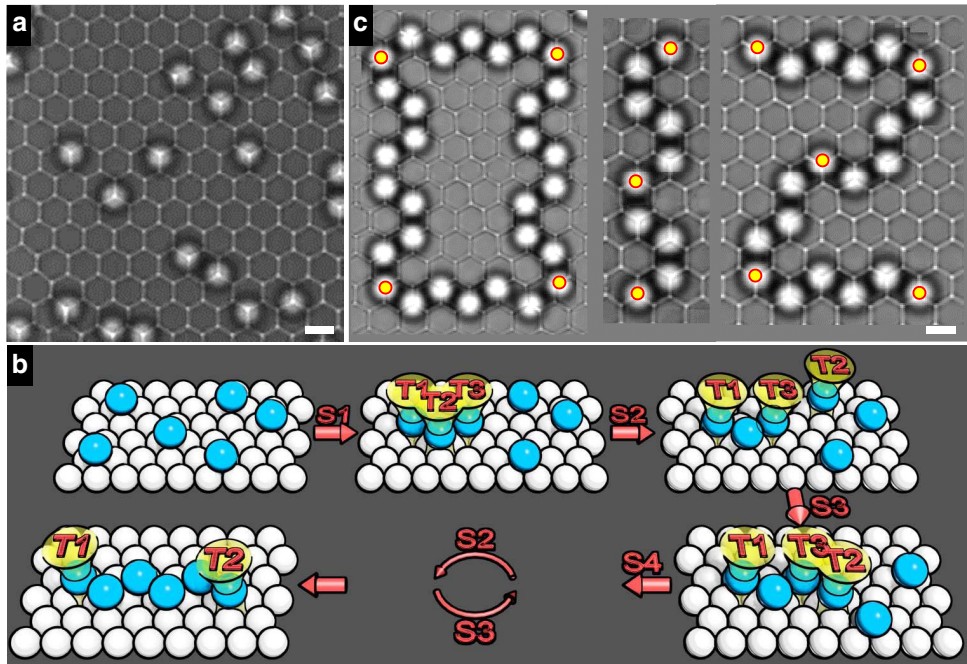

**Figure 2 | Vector assembly directed by optical tweezers.** (**a**) A typical image of the 2D colloidal system with the focal plane in-between the top particle layer and the bottom monolayer so that the monolayer appears as a uniform honeycomb lattice and the top particles as bright spots. Scale bar, 3 μm. (**b**) The iterative procedure of vector assembly, where tweezers 2 and 3 alternatively act as a shuttle to move wandering particles into position. The final chain is locked down by a mere two optical tweezers. T1 to T3 denote optical tweezers and S1 to S4 refer to assembly steps. (**c**) Vector structures of the digits 0, 1 and 2 with the yellow–red circles denoting optical tweezers. Scale bar, 3 μm.

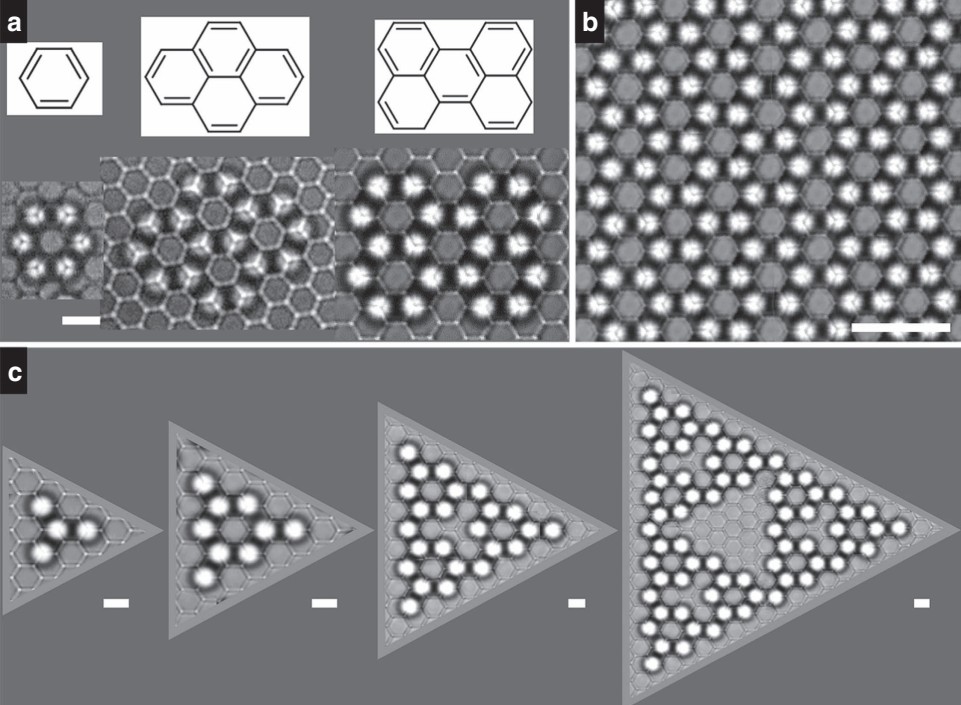

**Figure 3 | Colloidal arene structures.** (**a**) From left to right, colloidal benzene, pyrene and perylene. Scale bar, 3 μm. (**b**) Colloidene in analogy to graphene. Scale bar, 10 μm. (**c**) Four generations of a fractal honeycomb structure. Scale bar, 3 μm.

structures. Generally, more optical tweezers are required to lock down longer chains or larger structures. Two rules of thumb are that optical tweezers have to pin the ends and corners and that the number ratio of optical tweezers to particles is about 1/5. Following these rules and the iterative procedure, we can essentially build vector structures with arbitrary yet simple geometries, for example, the digits 0, 1 and 2 (Fig. 2c). In comparison with vector graphics, the operations like translation and rotation are not implemented in the present vector assembly. It is expected that chemical fixation of the as-formed structures could enable the implementation of these operations. In addition, we do not intend to parallel another feature of vector graphics, namely the infinite fidelity-conserved zoom-in or zoom-out, which in principle is not possible for any real materials.

**Colloidal arenes**. Recently, the analogy between colloids and atoms has led to insights into crystal nucleation/melting, glass transition and particle–particle interactions and spurred a growing interest in fabricating 'colloidal molecules'[20–23]. The most successful and popular fabrication approach so far is to decorate the particle surface with sticky patches of defined chemistry, size and location[8,21,23]. The patches endow the particles with bonding directionality and valence so that the colloidal atoms can assemble into simple colloidal molecules analogous to $H_2$, $CH_4$ and $CH_2 = CH_2$ (ref. 21). Fundamentally different from the patch approach that embeds the assembly information into the patches by complicated synthesis, the current vector approach encodes the assembly information into the substrate monolayer and a minimal number of optical tweezers. By doing so, we use simple spherical, undecorated

particles to construct a rich variety of 'colloidal arenes' that are inaccessible by the patchy approach.

The constructed 2D colloidal structures (Fig. 3) are finite honeycomb lattices in nature as a result of the hexagonal bottom monolayer and the top-to-bottom particle size ratio ∼1.1. These structures are morphologically analogous to aromatic compounds or arenes, for example, a 6-particle ring to benzene, a 16-particle structure to pyrene and a 20-particle structure to perylene (Fig. 3a). The zigzag chain in Fig. 1f is reminiscent of all-*trans* polyacetylene. These colloidal arenes are not fixed to the substrate; instead they are dynamic, fluctuating structures (Supplementary Movie 2). A single layer of colloidal honeycomb lattice as large as 50-by-50 μm is an analog of graphene and can thus be termed as 'colloidene' (Fig. 3b). The colloidene size is essentially limited by the domain size of the bottom monolayer and the operational area of the optical tweezers. Not only reported aromatic compounds can be replicated, but also abnormal, difficult-to-synthesize structures can be constructed at demand. Four generations of a fractal honeycomb structure (Fig. 3c) were built to demonstrate the power of the vector assembly strategy[24]. Moreover, it is possible to obtain free-standing colloidal arenes as a new class of bricks for hierarchical assembly of higher order structures in 3D if one can chemically fix the vector structures.

**Vector disassembly**. In the macroscopic world, people have developed various measures to break or disassemble things. For example, the entire bridges or towers collapse when their corner stones or foundations are demolished by explosion. A multistage space rocket can drop off its unnecessary parts stage-by-stage to

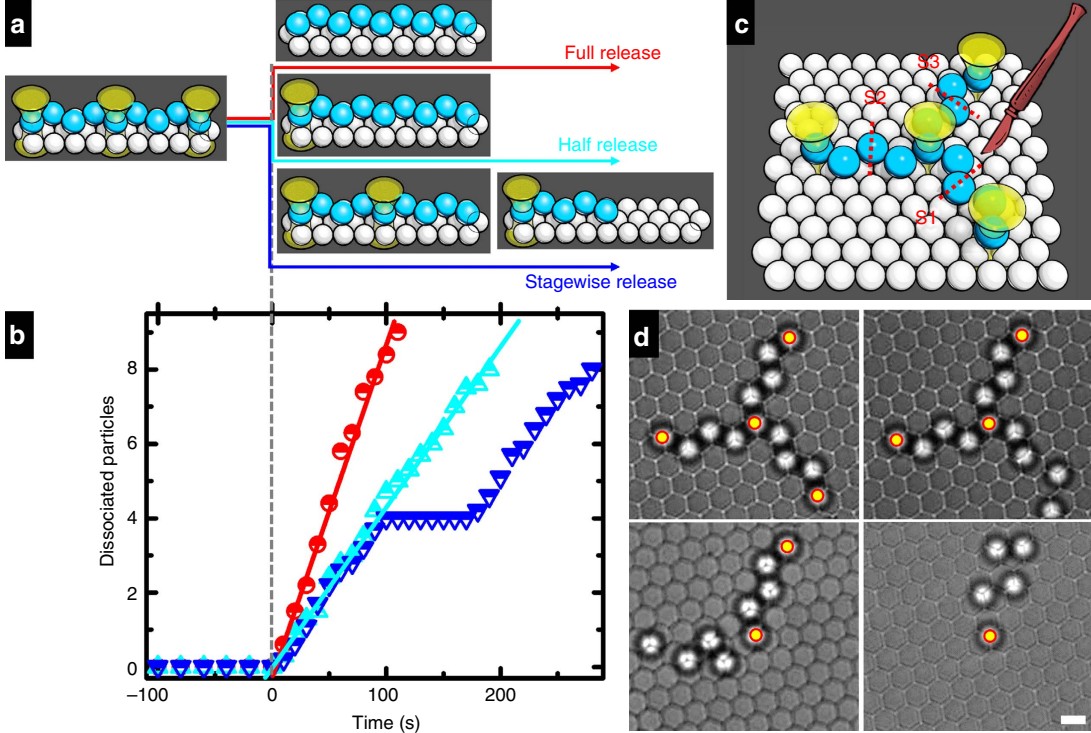

**Figure 4 | Two examples of vector disassembly.** (**a**) A scheme to disassemble a three-tweezer stabilized zigzag chain by unclicking the tweezers in three different manners. Full release, removing all the three tweezers at time 0 (red); half release, unclicking the end and middle tweezers at time 0 (cyan); and stagewise release, withdrawing the end tweezer at time 0 and middle tweezer at 160 s (blue). (**b**) The number of dissociated particles against time. Points are experimental data and straight lines linear fitting with colour coding the same as that of (**a**). The satisfactory linear fitting indicates that the particles detach from the chain end one by one at a statistically constant rate. The particle dissociation times are ∼12 and 23 s for the red and cyan lines, respectively. (**c**) A scheme to surgically cutoff a three-arm structure in three steps (S1 to S3). (**d**) Four snapshots of the disassembly process with yellow–red circles highlighting the operating optical tweezers. Scale bar, 3 μm.

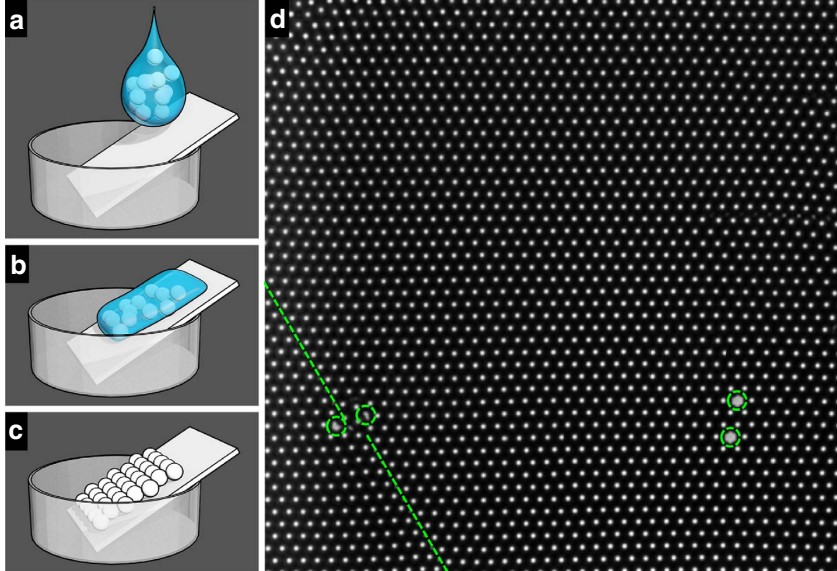

**Figure 5 | The colloidal monolayer.** (**a**–**c**) Scheme of monolayer fabrication. (**d**) A typical image of the as-prepared monolayer domain with point defects highlighted by green circles and line defects by green lines. The nearest white dot-to-white dot distance is 3 μm, the particle diameter.

reduce mass. A tissue or a body part, once corrupted beyond recovery, can be precisely cutoff in a surgery to prevent diseases from spreading. In the microscopic world, extensive efforts have been dedicated to spontaneous or directed assembly while much less attention has been paid to the reverse process, disassembly. The common practice is to make the assembly state of molecules or colloids unfavourable by a change of temperature, concentration or other environmental triggers, eventually causing the entire structure to dissolve or collapse[11,25,26]. Such practice clearly cannot parallel the flexibility and complexity of its macroscopic counterparts.

The colloidal structures in this work are held stable by pinning optical tweezers, the removal of which is expected to disassemble the structures by releasing the particles to free diffusion. In Fig. 4a, a 9-particle zigzag chain pinned by three optical tweezers was designed to be disassembled in three ways: full release, removing all three tweezers at 0 s; half release, removing one end and the middle tweezers at 0 s; and stagewise release, removing one end tweezer at 0 s and the middle tweezer at 160 s. The kinetics of chain dissociation were monitored by counting the number of particles that left their original positions (averaged over 10 parallel experiments, Fig. 4b). In the cases of full release and half release, the linear time dependence of the dissociation data (red and cyan points) indicates a stepwise depolymerization mechanism with one faster than the other. In the stagewise release, one end tweezer was removed at 0 s such that half of the chain is fully dissociated within 100 s. Then the mid tweezer was released at 160 s to trigger the disassembly of the remaining half chain, thus creating a 60-s suspension in the dissociation process (blue points). In Fig. 4c, three adjoined arms were designed to be surgically cutoff in sequence by removing the optical tweezers one after another clockwise. This surgical disassembly was followed by a time series of images (Fig. 4d). These two examples demonstrated the convenience, precision and flexibility of the disassembly triggered by tweezer-removal[27].

## Discussion

This paper has described the manifestation of the raster/vector concept originated from computer graphics in a 2D colloidal assembly system. Vector colloidal assembly was realized by manipulating particles on a bottom monolayer with optical

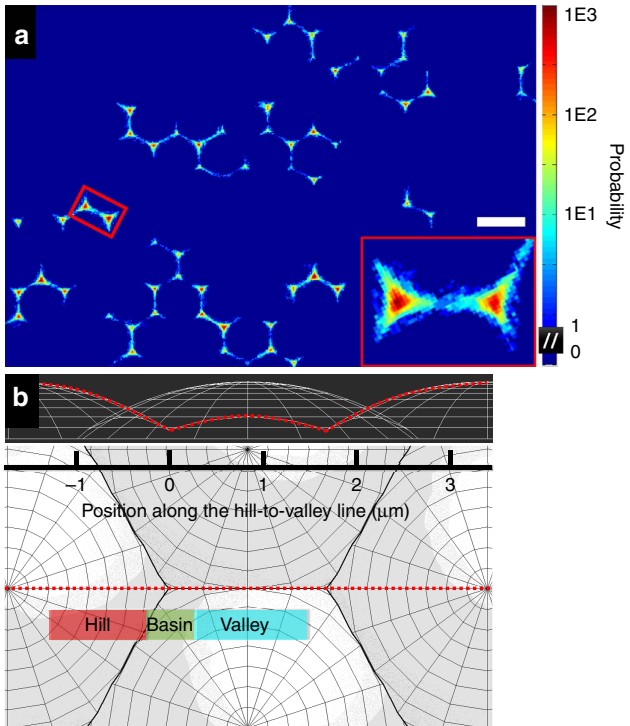

**Figure 6 | Particle diffusion on the colloidal monolayer.** (**a**) A positional probability map of multiple particles on the monolayer with a butterfly-like distribution enlarged in the insert. Scale bar, 3 μm. (**b**) Side and top views of the lowest surface that the top particle centre can explore. The red dash line is defined as the hill-to-valley line with hill, basin and valley regions indicated by different colours.

tweezers. Parallel to vector graphics, the current vector assembly features a minimal number of optical tweezers and transformations like elongation and shortening. We further used this vector approach to construct a collection of colloidal arenes including colloidenes that are not accessible by other methods, and demonstrated how the simple manipulations of optical tweezers

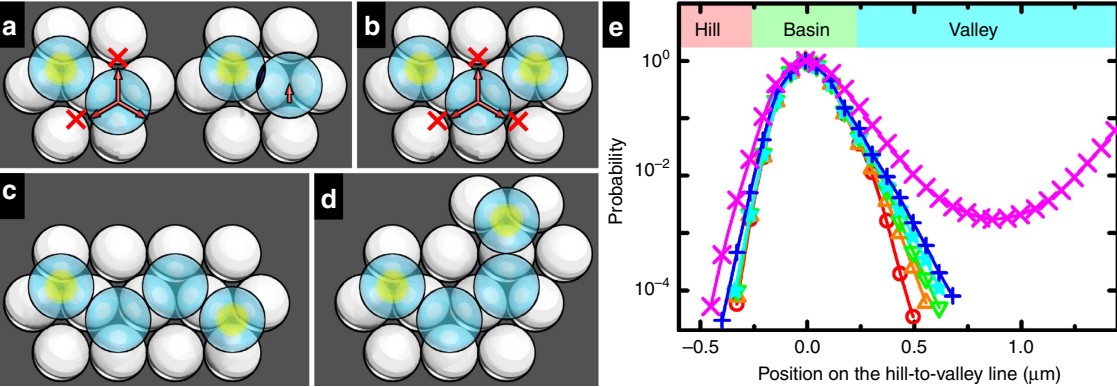

**Figure 7 | Stability of vector structures. (a–d)** 2-particle, 3-particle, *trans*-4-particle and *cis*-4-particle configurations with the optical tweezers as the yellow circles. Blocked hopping routes are highlighted in **a,b**. **(e)** Positional probability along the hill-to-valley line with red, orange, green, cyan, blue and magenta points corresponding to optical tweezer holding, 3-particle, *trans*-4-particle, *cis*-4-particle, 2-particle and free configurations, respectively. Side and top views of the lowest surface that the top particle centre can explore. The red dash line is defined as the hill-to-valley line with hill, basin and valley regions indicated by different colour.

can disassemble the vector structures in a surgical, flexible or stage-by-stage way.

We conclude with frank evaluation of the limitations in the present experiments and with optimistic outlooks into the future of vector assembly. First, the current vector assembly does not occur spontaneously but is directed by optical tweezers, so the vector structures cannot be produced in large quantity and 'click assembly' cannot be realized. The vector approach is, however, by no means limited to directed assembly. It is possible to mix a trace amount of carefully designed pin molecules with a large amount of dummy molecules so that molecular assemblies can be massively produced and be switched between micelles, vesicles and tubes upon simple clicks[28]. Second, chemical fixation of the vector structure is envisioned to be feasible given a number of well-established methods such as sintering, polymer grafting and cross-linking, and capillary condensation[29–31]. Once permanently fixed, the vector structures can be subject to translational or rotational operations or layer-by-layer stacking to construct 3D structures. Last but not least, the present assembly process is relatively time-consuming especially for large structures, but one can reduce consumption of time and labour by developing automated assembly approaches. Although those next steps are beyond the scope of this paper, we anticipate the vector assembly to open new doors for self-assembly and directed assembly.

## Methods

**Colloidal monolayer and sample cell.** A monolayer of 3 μm diameter silica particles (Tokuyama Inc.) was assembled on a No. 1 glass coverslip. In brief, one or two droplets of aqueous suspension of 4 wt% silica particles were dropped onto a piece of glass coverslip (pretreated by Piranha solution). The coverslip was gently tilted so that the suspension can quickly cover the coverslip and was subsequently dried by $N_2$, giving a colloidal monolayer (Fig. 5a–c). The particles were closely packed in a hexagonal lattice with minimum defects and single-crystal domains as large as 100-by-100 μm (Fig. 5d). The particles were completely fixed in the monolayer with no thermal or optical tweezer-induced motion at all. A sample cell was built on top of the colloidal monolayer and then filled by an aqueous suspension of dilute 3.4 μm diameter silica particles (Tokuyama Inc.), which will sediment to the bottom monolayer due to gravity. NaCl of 1 mM was added to screen any residual charges on the top and bottom silica particles so that the top particle-top particle and top particle-monolayer interactions are simple hard-core. The system was then ready to be observed by an inverted microscope. The top particles can be further manipulated by optical tweezers.

**Imaging and tracking.** Observations were made using an inverted optical microscope (Zeiss Observer Z1) with an EMCCD camera (Andor iXon). We visualized diffusion of the top particles using a × 100 oil objective focused between the top particle layer and the bottom monolayer so that the monolayer appears as a uniform honeycomb lattice and the top particles as bright spots[32]. Video images were collected typically at 10–100 fps for 2,000–4,000 frames, then analysed.

The particle tracking codes we used are standard in the colloid field (http://site.physics.georgetown.edu/matlab/) with refinements described in earlier publications from this research group[33,34]. In each movie, the centre of each particle was located in each time frame with 10 nm precision.

**Restricted Brownian motion of spherical particles on a colloidal monolayer.**
A positional probability map of multiple particles (Fig. 6a) was constructed from a 4,000 frame, 200 s long movie. The diffusion is clearly highly restricted to the honeycomb lattice defined by the bottom monolayer. A closer look into the unit cells reveals a prevailing butterfly-like distribution (Fig. 6a, insert). This distribution can be understood as the bottom monolayer provides a periodic gravitational-energy landscape for the top particles to diffuse on. The lowest surface that the top particle centre can explore is plotted in side and top views (Fig. 6b), where a medial line is highlighted as the 'hill-to-valley' line. The positional probability along this line is averaged over numerous unit cells and plotted in Fig. 7b (magenta points). The probability peaks in the basin, drops steeply in the hill side, and decreases moderately along the valley and eventually rises gradually at the other side of the valley. It is important to notice that the top particles can hop to nearby basins only through valleys but not hills.

**Optical tweezers.** A Mai Tai Ti-sapphire laser (Spectra-Physics) at wavelength of 800 nm was used to form the optical traps. The beam was first cleaned by a spatial filter set-up, then collimated and magnified to fill the operating panel of a Hamamatsu Spatial Light Modulator (LCOS-SLM) at a refresh rate of 60 Hz. The first-order beam was directed to the back aperture of the microscope and focused using oil-immersion × 100 objective (NA = 1.40) to form optical traps. By manipulating the phase field on the active component of the SLM with a modified open-source software[35], optical traps can be created and moved with ~1 μm precision with mouse inputs. The experiments are conducted at a reduced power to avoid heat generation.

**Stability of the vector structures.** We first consider two adjacent particles with one held tight to its basin by an optical tweezer and the other free to diffuse (Fig. 7a, left). Two hopping routes of the latter are blocked by the former unless the two hard particles unphysically overlap (Fig. 7a, right). In a three-particle configuration in Fig. 7b, two tweezer-holding neighbours prohibit the middle particle from hopping through any of the three routes. As shown in Fig. 7c,d, particles in 4-particle or longer chains are interlocked by mutual blockage. Such blockage is evidenced by the positional probability along the hill-to-valley line (Fig. 7e). The data for tweezer-holding particles in the basins and free particles are listed for comparison. While the probability in the hill and basin is more or less the same in different cases, the probability in the valley changes gradually with tightest blockage by the tweezer (red points), less-tight blockage by two tweezer-holding neighbours (orange, green and cyan points), and loose blockage by only one tweezer-holding neighbour (blue points).

It is interesting to notice that the *trans*-isomer of the 4-particle chains is more stable than the *cis* isomer (Fig. 7c,d, green and cyan points in Fig. 7e), reminiscent of the double bonds in chemistry. Longer chains tend to be less stable. Two optical tweezers can stabilize a zigzag chain up to about 10 particles, while more tweezers are required for longer chains or larger structures. Two conditions were found to be critical for the interlocked structures to be stabilized. One is that the top and bottom particles are of intermediate sizes so that the top particles can escape the basins by crossing the valleys but not the hills. For example, if the top particles are too small to be confined to the basin-valley regimes, they can easily diffuse over the

bottom monolayer despite the pinning optical tweezers. To extend our method to smaller particles (500 nm to 2 μm), one may use heavy particles such as $Fe_3O_4$ particles to enhance the gravitational confinement. The other is that the top/bottom particle size ratio is about ∼1.1 so that two neighbour top particles can occupy the *para*-basins and can hinder each other from moving to the *ortho*-basins.

**Data availability.** The authors declare that the data supporting the findings of this study are available within the article and its Supplementary Information Files.

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

## Acknowledgements

L.J. acknowledges support by the startup funding from Jinan University. S.G. acknowledges support by the Institute for Basic Science, project code IBS-R020-D1.

## Author contributions

L.J. and S.G. initiated the project and conceived the experiment. S.Y. and L.J. performed the experiments. B.T. set-up the optical tweezers. All authors contributed to analysing the results and writing the paper.

## Additional information

**Competing interests:** The authors declare no competing financial interests.

**Publisher's note**: 

