## [Peer Review File · Nature Communications]

REVIEWERS' COMMENTS:

Reviewer #1 (Remarks to the Author):

This is an interesting manuscript that draws inspiration from computer graphics to further develop colloidal assembly approaches. The paper is clearly written, the literature prior to this work is well cited. The potential impact of this paper, if published by Nature Communications is expected to be high. The authors critically assess the limitations of the demonstrated approach, though they are right that some of these limitations can be dealt with in future studies. In general, I think the paper will be of broad interest to the readership of Nature Communications and recommend it for being published in the present form.

Reviewer #2 (Remarks to the Author):

What an interesting analogy: from raster and vector graphics to colloidal assembly on a substrate. The authors discuss how to design colloidal patterns on a substrate using multiple optical tweezers. The variety of patterns they can form with this simple setup is quite remarkable, but more importantly this work provides a fresh and very intriguing new perspective in this field. This paper certainly meets the bar in creativity and relevance expected from a Nature Communications paper. I recommend it for publication as is.

Reviewer #3 (Remarks to the Author):

The following review relates to the manuscript of Yang et al. entitled "A Case Of Vector Assembly: Colloidal Assemblies and Click Disassembly". The work describes a method of optical tweezers assisted assembly of colloidal structures on a Langmuir-Blodgett assembled colloidal substrate. The main advantage here is that the optical tweezers are only required at specific pinning points in order to maintain ordering in large scale structures. This is achieved by imposing order through the underlying colloidal substrate and as such, relaxes the power requirement of optical trap multiplexing, which would otherwise require individual objects to be manipulated independently. The authors demonstrate the utility of the method by forming a number of arrangements that mimic molecular systems that could be used to dynamic simulations. Finally the authors discuss the benefits of using the above approach for deconstructing assemblies using the tweezers by removing judiciously chosen particles.

The novelty of the work is high and for particular problems the approach may serve to extend the capabilities of optical tweezers in the area of large-scale assembly. The underlying substrate does impose a strong symmetry requirement that means many assembled structures and as such it loses the flexibility of direct optical manipulation. I have strong doubts that this approach will not scale well with particle dimensions. For example, much smaller particle sizes have significant Brownian motion and it is questionable whether the hard-sphere interactions with the underlying substrate and adjacent will be sufficient to pin the colloids in place. I would like to see how well the technique works for smaller particles (e.g. 1 μ m or 0.5 μ m diameter spheres), which are more interesting for simulating condensed matter phenomena. I find that the need for the underlying colloidal monolayer and fixed nature of the assembled structures provides limited opportunities for post assembly optical manipulation. Finally, I find that there is an excessive use of gimmickry used to sell the science, which is not necessary. The vector graphics analogy I can appreciate, but it is a little superficial in its relation – the click disassembly analogue is extremely tenuous.

In summary I find this a nice piece of work and a novel idea definitely worthy of publication, but I feel it will be of limited impact.

Final Revision Requested for Manuscript NCOMMS-17-04501

Vector assembly of colloids on monolayer substrates

Response to Reviewer # 1

We thank the reviewer, who appreciated the analogue to computer graphics and recommended publication in its present form.

Response to Reviewer # 2

We thank the reviewer, who appreciated the new perspective provided by this work and recommended publication as is.

Response to Reviewer # 3

We thank the reviewer, who appreciated the novelty of this work and recommended publication. We have done our best, in the revised manuscript, to address the reviewer's concerns.

Concern about the applicability of the current method to smaller particles.

Indeed, the top and bottom particles are of intermediate sizes so that the top particles can escape the basins by crossing the valleys but not the hills. For example, if the top particles are too small to be confined to the basin-valley regimes, they can easily diffuse over the bottom monolayer despite the pinning optical tweezers. To extend our method to smaller particles (500 nm to 2 μm), one may use heavy particles such as Fe_3O_4 particles to enhance the gravitational confinement.

This paragraph of discussion is now added to the last paragraph in Methods section in the revised manuscript.

Concern about the analogue of disassembly to "click chemistry".

We agree this analogue is not necessary and therefore rename the subsection title into "vector disassembly". And we also omit all the analogues to "unclick" in the revised manuscript.